# Distinct and evolutionary conserved structural features of the human nuclear exosome complex

Piotr Gerlach[†], Jan M Schuller[†], Fabien Bonneau, Jérôme Basquin, Peter Reichelt, Sebastian Falk, Elena Conti*

Department of Structural Cell Biology, Max Planck Institute of Biochemistry, Munich, Germany

**Abstract** The nuclear RNA exosome complex mediates the processing of structured RNAs and the decay of aberrant non-coding RNAs, an important function particularly in human cells. Most mechanistic studies to date have focused on the yeast system. Here, we reconstituted and studied the properties of a recombinant 14-subunit human nuclear exosome complex. In biochemical assays, the human exosome embeds a longer RNA channel than its yeast counterpart. The 3.8 Å resolution cryo-EM structure of the core complex bound to a single-stranded RNA reveals that the RNA channel path is formed by two distinct features of the hDIS3 exoribonuclease: an open conformation and a domain organization more similar to bacterial RNase II than to yeast Rrp44. The cryo-EM structure of the holo-complex shows how obligate nuclear cofactors position the hMTR4 helicase at the entrance of the core complex, suggesting a striking structural conservation from lower to higher eukaryotes.

DOI: https://doi.org/10.7554/eLife.38686.001

*For correspondence:
conti@biochem.mpg.de

[†]These authors contributed equally to this work

Competing interests: The authors declare that no competing interests exist.

## Introduction

The eukaryotic RNA exosome is a conserved and versatile ribonuclease complex involved in many RNA quality-control and turnover pathways in both nuclear and cytoplasmic compartments. Besides eliminating defective and superfluous transcripts, the exosome has also processing functions in the maturation of nuclear RNA precursors, such as ribosomal RNAs (rRNAs) and small nucleolar RNAs (snoRNAs) (reviewed in *Chlebowski et al., 2013*; *Zinder and Lima, 2017*). Indeed, the exosome was originally discovered in *S. cerevisiae* as a complex of ribosomal RNA processing (Rrp) factors (*Mitchell et al., 1997*). Soon afterwards, it became apparent that the yeast exosome has a human counterpart in the so-called PM-Scl complex, which had been identified as the target of autoantibodies in patients suffering from polymyositis-scleroderma overlap syndrome (PM-Scl) (*Allmang et al., 1999b*). Several Mendelian diseases are now known to be associated with mutations in subunits of the human exosome complex (reviewed in *Morton et al., 2018*; *Reis et al., 2013*). While the yeast exosome has been extensively studied in the past two decades, mechanistic studies on the human complex have generally lagged behind.

Both the yeast and human exosomes are centered around a scaffold of nine catalytically inactive subunits forming a barrel-like structure, with a ring of six 'base' proteins and a ring of three 'cap' proteins (*Dziembowski et al., 2007*; *Liu et al., 2006*; *Makino et al., 2013*). The essential catalytic activity of the *S. cerevisiae* complex is contributed by the tenth subunit, an RNase II-like 3′–5′ exoribonuclease known as Rrp44 or Dis3 (*Dziembowski et al., 2007*). Yeast Rrp44 (yRrp44) is tethered to the base of the exosome barrel to form yExo-10 (*Bonneau et al., 2009*; *Makino et al., 2013*), the processive ribonuclease core common to the nuclear and cytoplasmic yeast exosome complexes. A long RNA-binding channel spans yExo-10, starting with a narrow entry pore in the cap ring,

continuing in the central cavity of the base ring and ending at the yRrp44 exoribonuclease site (*Bonneau et al., 2009*; *Makino et al., 2013*). Proteomic studies of human exosome complexes recently showed that the 9-subunit barrel associates with two distinct yRrp44 orthologues, hDIS3 and hDIS3L, to form nucleoplasmic and cytoplasmic hEXO-10 complexes, respectively (*Staals et al., 2010*; *Tomecki et al., 2010*). The third paralogue of yRrp44, the cytoplasmic protein hDIS3L2, instead functions independently of the RNA exosome (*Chang et al., 2013*; *Lubas et al., 2013*; *Malecki et al., 2013*). Intriguingly, mutations in hDIS3 but not hDIS3L have been identified in patients with multiple myeloma (*Chapman et al., 2011*; *Lionetti et al., 2015*; *Tomecki et al., 2014*), underscoring the importance of the nuclear form of the complex.

The nuclear exosome includes four specific cofactors that are conserved from lower to higher eukaryotes. In yeast, the yRrp6 exoribonuclease and its binding partner yRrp47 are tethered with high affinity to the cap of yExo-9 (*Kowalinski et al., 2016*; *Makino et al., 2015*; *Zinder et al., 2016*) and target the complex to the nuclear compartment (*Gonzales-Zubiate et al., 2017*). The yRrp6-yRrp47 dimer together with another stably associated subunit, yMpp6, recruits the essential 3′−5′ RNA helicase yMtr4 (*Falk et al., 2017*; *Schuch et al., 2014*; *Wasmuth et al., 2017*). It has long been suspected that yMtr4 helps to unwind RNA substrates and to present them to the processive core of the exosome. Indeed, in a recent cryo-EM structure of the yeast 14-subunit nuclear exosome bound to a pre-60S ribosomal particle, we could observe the physiological RNA substrate (a 5.8S pre-ribosomal RNA precursor) being channeled from yMtr4 into the exosome core (*Schuller et al., 2018*). Orthologues of these nuclear exosome cofactors have been identified in human cells: EXOSC10 (hRRP6), C1D (hRRP47), MPH6 (hMPP6) and MTREX (hMTR4) (*Schilders et al., 2007a*; *2005*; *Schilders et al., 2007b*).

The human nuclear exosome shares similar functions with the yeast complex, but also shows important functional differences (*Kilchert et al., 2016*; *Ogami et al., 2018*; *Sloan et al., 2012*). The processing of the 5.8S rRNA is a common role of nucleoplasmic exosome complexes, with similar intermediates being formed in yeast and human cells (*Allmang et al., 1999a*; *Briggs et al., 1998*; *Tafforeau et al., 2013*). Degradation of the 5′ ETS, a byproduct of ribosome biogenesis, is also a function of both yeast and human nucleolar exosome complexes, but different cofactors are involved (*Sudo et al., 2016*). The major function of the human nuclear exosome, however, appears to lie in quality control pathways that counteract pervasive transcription initiation and defective transcription termination (*Belair et al., 2018*; *Ogami et al., 2018*; *Szczepińska et al., 2015*). For example, the nucleoplasmic exosome targets promoter upstream transcripts (PROMPTs) that arise due to anti-sense transcription from bidirectional promoters (*Preker et al., 2008*) and prematurely terminated products of protein coding genes (*Szczepińska et al., 2015*). In line with the prevalence of these pathways, human cells have evolved specialized cofactor complexes built around hMTR4 (*Lubas et al., 2011*; *Meola et al., 2016*; *Ogami et al., 2018*). Here, we report a mechanistic comparison of the human and yeast nuclear exosomes with the aim to understand the extent of their evolutionary conservation and to identify species-specific features.

## Results and discussion

### The RNA path in the nuclear exosome core complex is longer in human than in yeast

We set out to compare the RNA-binding properties of yeast yExo-10 and human nuclear hEXO-10. Since the structure of the yeast exosome core has been discussed in several publications (*Falk et al., 2017*; *Kowalinski et al., 2016*; *Liu et al., 2006*; *Makino et al., 2013*; *Wasmuth et al., 2014*), we will refer to human exosome components (EXOSC) of the 9-subunit barrel with the corresponding names from *S. cerevisiae*, namely hCSL4 = EXOSC1, hRRP4 = EXOSC2, hRRP40 = EXOSC3, hRRP41 = EXOSC4, hRRP46 = EXOSC5, hMTR3 = EXOSC6, hRRP42 = EXOSC7, hRRP43 = EXOSC8 and hRRP45 = EXOSC9. We engineered a catalytically inactive mutant of human hDIS3 ($hDIS3_{cat}$ Asp146Asn, Asp487Asn) analogous to the previously characterized yRrp44 mutant ($yRrp44_{cat}$ Asp171Asn, Asp551Asn [*Bonneau et al., 2009*]). We reconstituted the corresponding recombinant mutant complexes ($yExo-10_{cat}$ and $hEXO-10_{cat}$) (*Figure 1A*) and carried out RNase protection assays to determine their RNA-binding footprint (*Figure 1B*). In these assays, a body-labeled single-stranded $(CU)_{48}C$ RNA was incubated with the catalytically inactive exosome complexes and treated

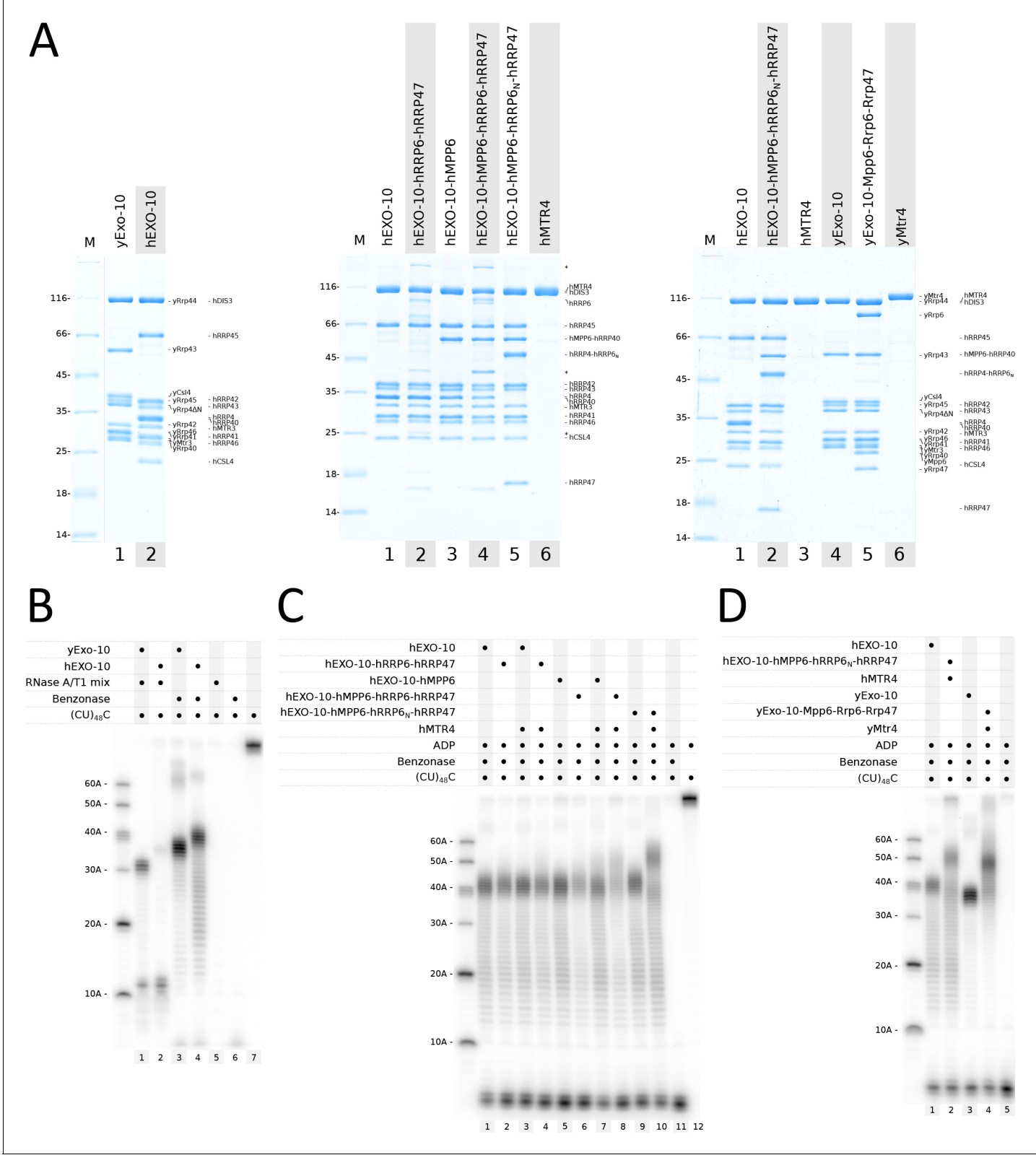

**Figure 1.** Biochemical analysis of the RNA-binding paths in yeast and human nuclear exosome complexes. (**A**) Coomassie-stained 12.5% SDS-PAGE gels showing reconstituted yeast and human exosome complexes and subunits used in the assays in panels (**B–D**). All samples correspond to the pooled peak fractions from size exclusion chromatography. hMPP6-hRRP40 and hRRP4-hRRP6$_N$ indicate the genetically linked fusion proteins, as described in the text. Undefined contaminants and degradation products from the hRRP6-hRRP47 preparation are indicated with asterisks.(**B–D**) RNase

*Figure 1 continued on next page*

*Figure 1 continued*

protection assays showing the RNA fragments obtained upon RNase treatment with $^{32}$P body-labeled (CU)$_{48}$C 97-mer RNA in the presence of the indicated protein complexes. After incubation with RNase A/T1 or with benzonase (*Serratia marcescens* endonuclease), the reactions products were analyzed by electrophoresis on a 12% acrylamide and 7M urea gel, followed by phosphorimaging. Protein concentrations were 1 μM (panel B) or 500 nM (panels C, D). Substrate concentration was 250 nM. The outer left lanes were loaded with size markers. Note that the size of fragments obtained in these in vitro assays is not an absolute measure of the length of the RNA-binding channel but is relative to the RNase used (panel B, compare lanes 1 and 3).

DOI: https://doi.org/10.7554/eLife.38686.002

with endoribonucleases, which digest accessible solvent-exposed portions of the RNA. Initially, we used an RNase A and RNase T1 endonuclease mixture. As we had previously reported (*Bonneau et al., 2009*), yeast yExo-10$_{cat}$ showed a bimodal protection pattern in RNase A/T1 assays, with the accumulation of 31–33 and 11–12 nucleotide fragments (*Figure 1B*, lane 1). The long 31–33 nucleotide fragments correspond to the so-called channel path, whereby RNA traverses the central channel of yExo-9 to reach the exoribonuclease site of yRrp44 (*Makino et al., 2013*). This path is used by the majority of RNAs in yeast (*Drazkowska et al., 2013*; *Schneider et al., 2012*; *Wasmuth and Lima, 2012*). The short 11–12 nucleotide fragments are thought to reflect the presence of an alternative direct path to the exoribonuclease site that is used in vivo by a limited number of nuclear RNAs (*Han and van Hoof, 2016*). The two RNA paths have been visualized structurally, and correlate with two different conformations of the yRrp44 ribonuclease: a closed conformation for the long channel path and an open conformation for the short direct path (*Makino et al., 2015*, *2013*).

In the case of human hEXO-10$_{cat}$, treatment with RNase A/T1 also resulted in a bimodal distribution with the accumulation of short fragments (11–12 nucleotides) and long fragments (~34–37 nucleotides), but the latter were much less abundant as compared to yExo-10$_{cat}$ (*Figure 1B*, lane 2). We changed assay conditions by using benzonase, a *Serratia marcescens* endonuclease with broad substrate specificity. Benzonase treatment resulted in a unimodal protection pattern for both the yeast and the human complexes, with only the long fragments accumulating (*Figure 1B*, lanes 3 and 4). The absence of short fragments suggested that the channel path predominates with the single-stranded RNA substrate used and that a portion of the RNA between the 9-subunit barrel and the ribonuclease may be accessible to small endoribonucleases (e.g. RNase A/T1, ~14 kD and ~11 kDa) but not to larger ones (e.g. the homodimeric benzonase, ~60 kDa). The long fragments accumulating upon benzonase treatment spanned 35–39 nucleotides for yeast yExo-10$_{cat}$ and 39–43 nucleotides for human hEXO-10$_{cat}$ (*Figure 1B*, lanes 3 and 4). We concluded that the human nuclear exosome core is likely to thread RNA through the central channel, but the path is more extended than in the yeast complex. The reason for this difference was unclear, considering that the yeast yExo-9 and human hEXO-9 barrels have a similar overall size (*Liu et al., 2006*; *Makino et al., 2013*) and that yRrp44 and hDIS3 are expected to be structural homologs.

## The human nuclear exosome cofactors extend the RNA channel of the core complex

Next, we tested the effect of the nuclear cofactors. In case of the yeast proteins, we have previously shown biochemically that the size of the protected RNA fragments increases of about 10–15 nucleotides when yExo10$_{cat}$ is incubated with its nuclear cofactors, including a catalytically inactive version of yRrp6 (yRrp6$_{cat}$ Asp296Asn), yRrp47, yMpp6 and yMtr4 in the presence of ADP (*Falk et al., 2017*). Structural studies have shown that this footprint reflects the channeling of RNA from yMtr4 into the exosome core (*Schuller et al., 2018*). We purified full-length hMTR4 from bacterial cell expression and the corresponding catalytically inactive hRRP6$_{cat}$-hRRP47 complex (hRRP6$_{cat}$ Asp371Asn) using mammalian cell expression from stably transfected HEK293T cells. In the case of hMPP6, we could not obtain homogenous samples when expressing this small low-complexity protein in isolation. Based on the structural information from the yeast yMpp6-exosome structures (PDB 5OKZ and 5VZJ) (*Falk et al., 2017*; *Wasmuth et al., 2017*), we genetically linked the C-terminus of hMPP6 to the N-terminus of the cap protein hRRP40 and reconstituted the corresponding exosome core complex containing the hMPP6-hRRP40 fusion protein (*Figure 1A*, central panel).

We added different subsets of nuclear cofactors in RNase protection assays with benzonase. The covalently linked hEXO-10$_{cat}$-hMPP6 complex behaved as wild-type hEXO-10$_{cat}$ (*Figure 1C*, compare lane 1 and 5). When incubating hEXO-10$_{cat}$-hMPP6 with hRRP6$_{cat}$-hRRP47 and hMTR4 in the presence of ADP, we observed a shift to longer fragments (*Figure 1C*, lane 8). Incubating hEXO-10$_{cat}$ and hMTR4 in the presence of either hRRP6$_{cat}$-hRRP47 (*Figure 1C* in lane 4) or hMPP6 (lane 7) showed only a modest shift to longer species (~50 nucleotides). The presence of all four cofactors and ADP was required to have a stronger shift to ~50 nucleotides (lane 8). Thus, it appears that the Mtr4-exosome needs at least one cofactor to be stabilized and all of them to show strong association, both in the yeast and human system.

In the cryo-EM structure of yeast nuclear yExo-14 with a pre-60S substrate (*Schuller et al., 2018*), we had observed that channeling through yMtr4 requires a large conformational change of yRrp6-yRrp47 (*Makino et al., 2015*): the ribonuclease domain of yRrp6 was displaced from the top of yExo-9 (to allow binding of the yMtr4 helicase on the yRrp4 cap protein) and the N-terminal hetero-dimerization module (yRrp6$_N$-yRrp47) was displaced from the yRrp6 ribonuclease domain (and instead bound yMtr4). The prediction from this observation is that channeling through the helicase does not require (and may actually be competing with) the ribonuclease domain of yRrp6. We tested whether the ribonuclease domain of hRRP6 is also dispensable for channeling RNA through hMTR4 by assaying the hRRP6$_N$-hRRP47 hetero-dimerization module alone. In a wild-type situation, both the N-terminal hetero-dimerization module and the central ribonuclease domain of yRrp6 are tethered to the exosome core via the high-affinity interaction of the yRrp6 C-terminal domain with the cap protein yCsl4 (*Kowalinski et al., 2016*; *Makino et al., 2013*). To test the influence of hRRP6$_N$-hRRP47 alone, we tethered it to the exosome core using a similar strategy described above for hMPP6. Based on the cryo-EM structural information of the yeast complex (PDB 6FSZ) (*Schuller et al., 2018*), we genetically linked the N-terminus of hRRP6$_N$ to the C-terminus of hRRP4 and reconstituted the corresponding exosome core complex containing the hMPP6-hRRP40 and hRRP4-hRRP6$_N$ fusion proteins (*Figure 1A*, central panel). In RNase protection assays, the hEXO-10$_{cat}$-hMPP6-hRRP6$_N$-hRRP47 complex resulted in a similar pattern as hEXO-10$_{cat}$ (*Figure 1C*, lanes 9 and 1). Upon incubation with hMTR4 and ADP we observed a defined shift to longer fragments (*Figure 1C*, lane 10). Using covalently linked hRRP6$_N$ instead of f.l. hRRP6 appeared to even stabilize the 50 nucleotide fragments (compare lanes 8 and 10). The most likely reason for this stabilization is the improved biochemical properties and stability of hRRP6$_N$ as compared to the full-length protein, allowing us to overcome the hRRP6 stoichiometry issues in the reconstituted complexes (*Figure 1A*). When comparing the 10-subunit core complexes with the 14-subunit nuclear holo-complexes, we observed a similar increase in the size of the protected fragments in both the yeast and human systems (*Figure 1D*).

## Cryo-electron microscopy of a human nuclear exosome

Since the covalently linked hEXO-10$_{cat}$-hMPP6-hRRP6$_N$ complex bound to hRRP47 and hMTR4 (hereafter referred to as hEXO-14$_{cat}$) behaved in vitro as a stable and functional form of the nuclear exosome, we proceeded to analyze its structure using cryo-electron microscopy. We purified hEXO-14$_{cat}$ in large scale to homogeneity, incubated with Mg-ADP and a single stranded RNA (U$_{44}$) and subjected it to mild crosslinking with BS3. We used this sample to collect more than 8000 micrographs (2.5 million particles) on a Titan Krios equipped with a K2 camera. Upon 2D classification and initial 3D classification, we observed two distinct particle populations that were then subjected separately to another round of 3D classification. From the larger particle population we obtained the structure of human hEXO-10$_{cat}$-hMPP6 at 3.80 Å resolution and from a smaller one we obtained the structure of hEXO-14$_{cat}$ at 6.25 Å resolution (*Figure 2—figure supplements 1–3*).

## The human hEXO-9 and yeast yExo-9 scaffolds share extensive structural similarities

The human hEXO-10$_{cat}$-hMPP6 cryo-EM structure revealed unambiguous density for the hEXO-9 barrel and for the hDIS3 ribonuclease (*Figure 2*). We first fitted the atomic coordinates from the 3.3 Å crystal structure of human hEXO-9 that had been previously reported (*Liu et al., 2006*). The excellent quality of the cryo-EM density allowed us to significantly improve the stereochemistry of the hEXO-9 atomic model, which now has less than 1% Ramachandran and rotamer outliers (*Table 1*).

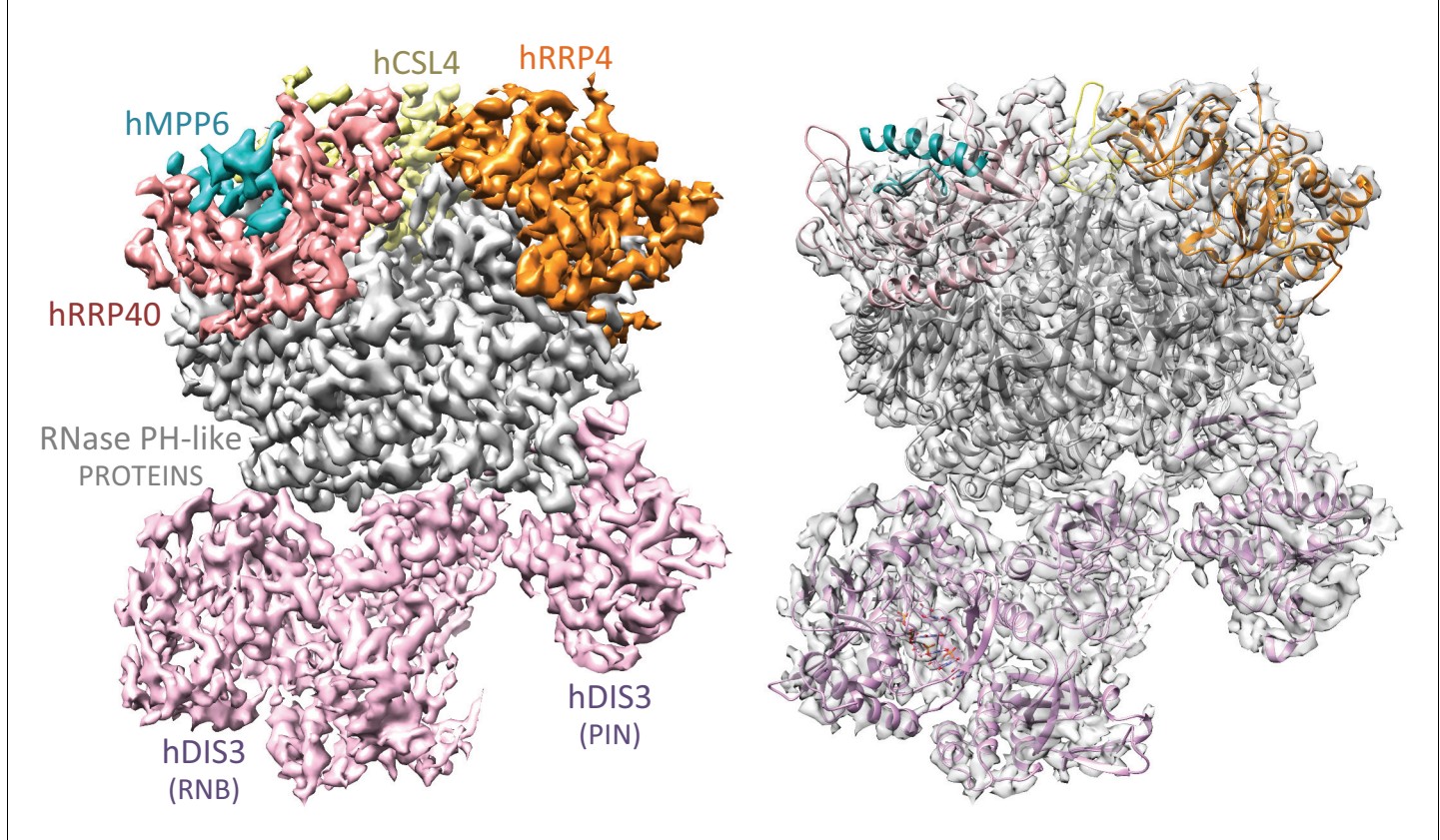

**Figure 2.** Cryo-EM structural analysis of a human nuclear exosome core. Cryo-EM surface representation (left panel) and cryo-EM density (right panel) of hEXO-10$_{cat}$-hMPP6 with the corresponding atomic coordinates of the individual exosome subunits. The hEXO-9 barrel comprises a base ring of 6 RNase PH-like subunits (all shown in gray) and a cap ring of 3 S1/KH-like proteins (shown at the top in yellow, orange and salmon). The PIN and exoribonuclease regions of hDIS3 are indicated (shown in light pink). The hMPP6 cofactor (shown in teal) is bound to the protein hRRP40.

DOI: https://doi.org/10.7554/eLife.38686.003

The following figure supplements are available for figure 2:

**Figure supplement 1.** Cryo-EM data collection and 2D classification.

DOI: https://doi.org/10.7554/eLife.38686.004

**Figure supplement 2.** Cryo-EM data processing scheme.

DOI: https://doi.org/10.7554/eLife.38686.005

**Figure supplement 3.** Cryo-EM data quality.

DOI: https://doi.org/10.7554/eLife.38686.006

The resulting structure of hEXO-9 has the characteristic base ring of adjacent hRRP41-hRRP42-hMTR3-hRRP43-hRRP46-hRRP45 subunits and the cap ring of adjacent hRRP4-hCSL4-hRRP40 subunits first observed by X-ray crystallography (*Liu et al., 2006*) (*Figure 3A*). In the cryo-EM structure, hCSL4 is only partially ordered. Similar observations have been made in a previous cryo-EM structure of yeast yExo-10 (*Wang et al., 2007*), as yCsl4 is known to be stabilized by the C-terminal region of yRrp6 (*Makino et al., 2013*).

The cryo-EM reconstruction showed several differences as compared to the X-ray structure (*Liu et al., 2006*). First, in the cryo-EM structure the C-terminal tail of hRRP45 (after residue 278) does not wrap around other subunits of the base. Instead, residues 280–287 bend backwards to form a β-strand, completing the hRRP45 β-sheet (*Figure 3B*). Second, the N-terminus of hMTR3 is well ordered as compared to the hEXO-9 crystal structure: it aligns on the hRRP43 β-sheet and then continues in the internal channel of hEXO-9 (*Figure 3C*). A similar arrangement was previously observed in the yeast yMtr3 orthologue (*Kowalinski et al., 2016*; *Makino et al., 2015*; *2013*; *Zinder et al., 2016*). The N-terminus of hRRP4 is also well ordered in the cryo-EM density and packs against the hRRP42 β-sheet (*Figure 3D*). An analogous interaction has been observed for

**Table 1.** Statistics of the hEXO-10$_{cat}$-hMPP6 cryo-EM atomic model.

|  | hEXO-10$_{cat}$-hMPP6 |
| --- | --- |
| Data collection |  |
| Microscope | Titan Krios |
| Camera | Gatan K2 Summit |
| Magnification | 105,000 x |
| Voltage (kV) | 300 |
| Electron dose (e⁻/Å²) | 46.9 |
| Dose rate (e⁻/pixel/s) | 8.55 |
| Defocus range (µm) | 0.5–3.5 |
| Pixel size | 1.35 |
| Reconstruction |  |
| Micrographs collected | 8047 |
| Particles in 3D classification | 691,785 |
| Particles in final refinement | 110,958 |
| Refinement |  |
| Resolution (Å) | 3.8 |
| No. atoms | 22117 |
| Protein | 21997 |
| RNA | 120 |
| Map sharpening *B*-factor | −153 |
| R.m.s deviations |  |
| Bond lengths (Å) | 0.006 |
| Bond angles (°) | 0.926 |
| Ramachandran plot |  |
| Favored (%) | 94.2 |
| Allowed (%) | 5.7 |
| Rotamers ouliers (%) | 0.65 |
| MapCC Global | 0.79 |

DOI: https://doi.org/10.7554/eLife.38686.010

yeast yRrp4 (*Kowalinski et al., 2016*; *Makino et al., 2015*, *2013*; *Zinder et al., 2016*). Thus, the yeast and human barrels are even more similar than previously thought. As a note, the cryo-EM reconstruction also showed density at the top of hRRP40 at the same position where yeast yMpp6 binds yExo-9 (*Falk et al., 2017*; *Schuller et al., 2018*; *Wasmuth et al., 2017*) (*Figure 3E*). In comparison to the yeast structure, the cryo-EM density of the human complex shows an additional structural feature of hMPP6, an α-helix that interacts intra-molecularly with an extended segment of hMPP6 (*Figure 3E*). We conclude that while the other cofactors of the nuclear exosome detached from the core in this particle population, the hMPP6 interaction with the core complex remained.

## Human hEXO-9 and yeast yExo-9 bind the exoribonuclease with different strengths

The hEXO-9 scaffold binds hDIS3 at the base of the barrel (*Figure 2*). hDIS3 has a similar domain organization as yeast yRrp44, with a PIN-domain region followed by an exoribonuclease region typical of the RNase II family of proteins (*Tomecki et al., 2010*). We first generated a homology model of hDIS3 based on the atomic coordinates of yeast yRrp44 (*Makino et al., 2013*). In the case of the PIN-domain region, the homology model fitted well in the EM density at the bottom of hRRP41 (*Figure 2*), necessitating only minor adjustments. Besides the classical PIN domain, this region includes an N-terminal segment known in the *S. cerevisiae* orthologue as the CR3 motif (*Schaeffer et al.,*

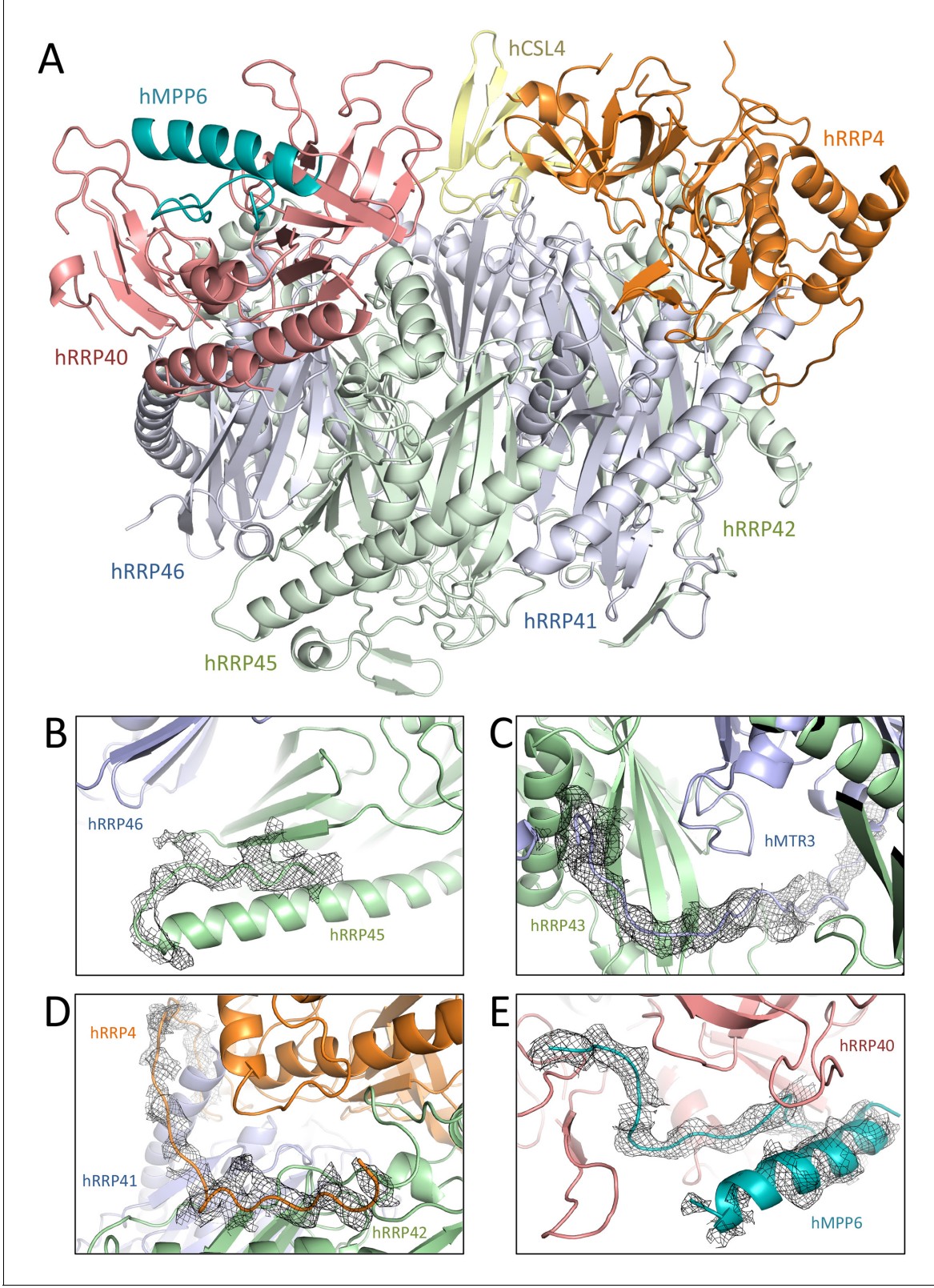

**Figure 3.** Human and yeast exosome cores: extensive similarities of the central scaffolds. (**A**) Structure of the hEXO-9 barrel from the cryo-EM reconstruction bound to hMPP6 (teal). In hEXO-9, the cap proteins are shown in the same colors as in panel A and the base proteins are in light blue (for the Rrp41-like proteins hRRP41, hRRP46 and hMTR3) and light green (for the Rrp42-like proteins hRRP42, hRRP43 and hRRP45). The definition of Rrp41-like and Rrp42-like follows the original description in (*Lorentzen et al., 2005*). The superposition of the human and yeast scaffolds is shown in

*Figure 3 continued*

*Figure 3—figure supplement 1.* (**B–D**) Zoom-ins at specific regions of human hEXO-9 described in text, with cryo-EM density superposed. (**E**) Zoom-in at hMPP6 fragment bound to hRRP40, with cryo-EM density superposed.

DOI: https://doi.org/10.7554/eLife.38686.007

The following figure supplements are available for figure 3:

**Figure supplement 1.** Similar scaffolds of yeast and human exosome cores.

DOI: https://doi.org/10.7554/eLife.38686.008

**Figure supplement 2.** Zoomed in views of the hEXO-9 subunits.

DOI: https://doi.org/10.7554/eLife.38686.009

*2012*). Similarly to the yeast exosome structure (*Makino et al., 2013*), the corresponding CR3 motif of hDIS3 forms a long β-hairpin wedged between hRRP41-hRRP42 (*Figure 4A*). However, the CR3 motif of hDIS3 is twenty-residue shorter and lacks a yRrp41-interacting loop (corresponding to yRrp44 residues 60–67, *Figure 4A*). Furthermore, the human complex lacks another interaction between the core and the exoribonuclease that was observed in the yeast exosome, namely the contact between yRrp44 and the C-terminus of yRrp45 (*Figure 4B*). Consistently, while yeast yRrp44 can stably interact with the yRrp41-yRrp45 heterodimer (*Bonneau et al., 2009*), size-exclusion chromatography experiments with recombinant proteins showed that the interaction between the human orthologues is weaker (*Figure 4C*).

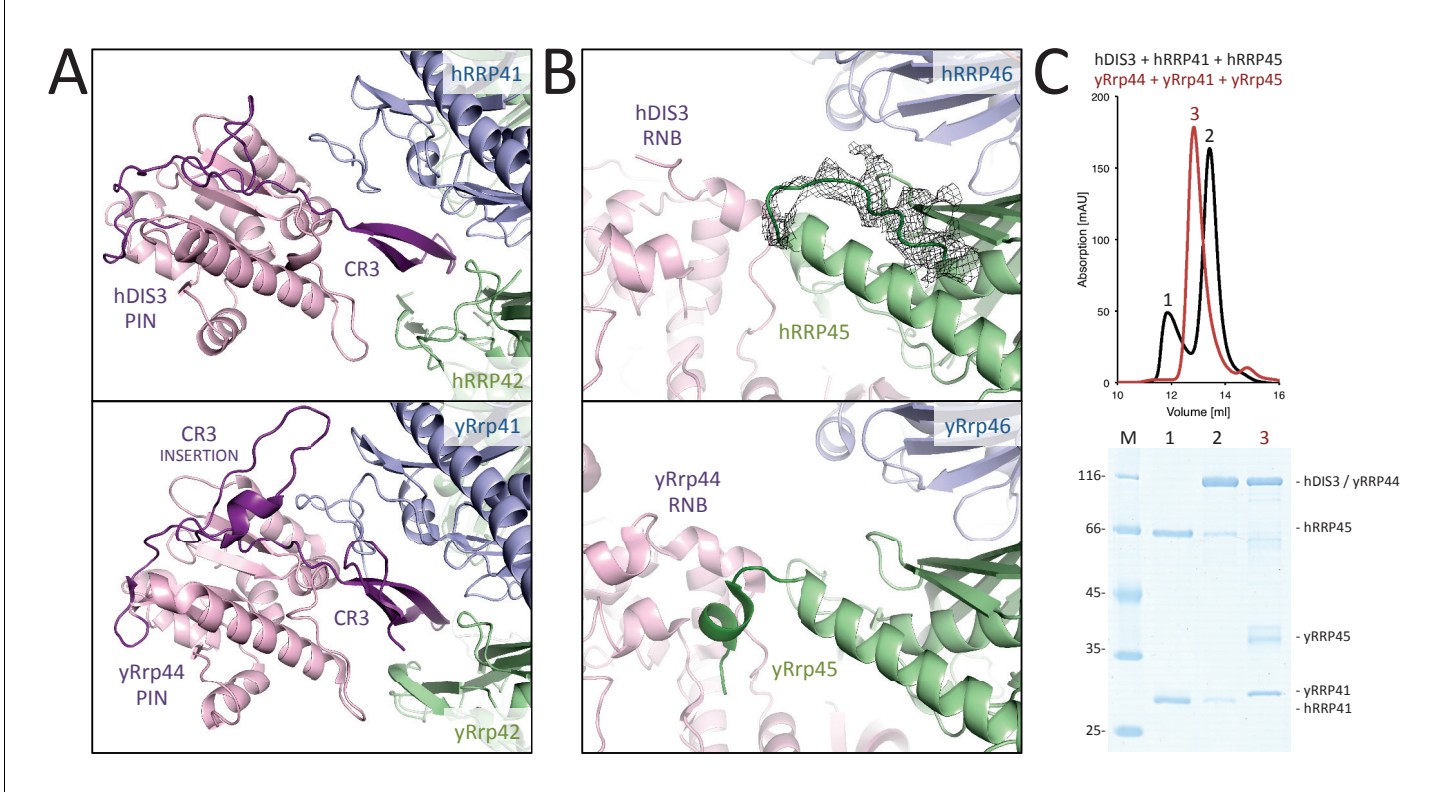

**Figure 4.** Human and yeast exosome cores: differential binding to the exoribonuclease. (**A-B**) Interactions made by the PIN region (**A**) and the RNB region (**B**) of the exoribonuclease (pink) with the proteins of the central scaffold (shown in the same colors as in *Figure 3*). The figures show the comparisons between the human cryo-EM structure (upper panels) and previous yeast X-ray structures (lower panels) (*Makino et al., 2015, 2013*). The human complex shows fewer interactions at the structural level as compared to the yeast complex. (**C**) Size exclusion chromatography experiments showing the weaker binding of hDIS3 to hRRP41-hRRP45 as compared to that of the yeast orthologues.

DOI: https://doi.org/10.7554/eLife.38686.011

## Human hDIS3 is structurally more similar to RNase II than to the yeast orthologue Rrp44

We then traced the hDIS3 exoribonuclease in the density adjacent to the PIN domain. The exoribonuclease region includes two N-terminal cold-shock domains (CSD1 and CSD2), a catalytic domain (RNB) and a C-terminal S1 domain (*Figure 5*). The three OB folds are positioned on top of the RNB domain, with CSD1 and CSD2 forming a lobe on one side of the RNB domain and facing the S1 domain on the other side (*Figure 5*). A similar overall arrangement has been described for all other known members of this protein family (*Frazão et al., 2006*; *Lorentzen et al., 2008*) (*Faehnle et al., 2014*) (*Figure 5*). However, there are distinct features in the precise arrangement of the CSD lobe. In hDIS3, the CSD lobe is separated from the S1 lobe by a large funnel-like cleft, reminiscent of OB fold arrangement present in RNase II and DIS3L2 (*Faehnle et al., 2014*; *Frazão et al., 2006*) (*Figure 5A,C,D*). In contrast, in yeast yRrp44 a rotation of the CSD lobe towards the S1 domain narrows the cleft (*Lorentzen et al., 2008*; *Makino et al., 2015*) (*Figure 5B*). Furthermore, two structural features of CSD1 specific to the yeast protein occlude and seal the narrow cleft (an extended segment at yRrp44 residues 327–347 and a helix at residues 366–384) (*Figure 5B,E*). As a result, yRrp44 does not feature an apical opening between the OB-fold domains. Instead, the rotation of the CSD lobe creates a lateral opening for the RNA, between the CSD1 and RNB domains (*Figure 5B*). In previous crystal structures of the RNase II family members, the position of the OB folds has been shown to shape the path with which RNA enters the catalytic chamber of the RNB domain: in yeast Rrp44, RNA enters the catalytic chamber of the RNB domain from the lateral opening between the CSD1 and RNB domains (*Lorentzen et al., 2008*) while in RNase II and DIS3L2, RNA enters from the apical opening between the CSD lobe and the S1 domain (*Faehnle et al., 2014*; *Frazão et al., 2006*) (*Figure 5B–D*). As described below, the RNA-binding path in hDIS3 is also determined by the position of the OB folds (*Figure 5A*).

## The RNA channel path in the human exosome core is underpinned by an open conformation of hDIS3

After the generation of accurate atomic models for all the human exosome core proteins, we analyzed the overall conformation of hEXO-10 and its RNA-binding path (*Figure 6*). In the cryo-EM structure, the overall position of the hDIS3 exoribonuclease region on hEXO-9 (*Figure 6A*) does not resemble the closed conformation with which yeast yRrp44 binds RNAs entering from the long channel path (*Makino et al., 2013*) (*Figure 6B*) but rather the open conformation with which yeast Rrp44 binds RNA in the direct access path (*Makino et al., 2015*) (*Figure 6C*). The PIN region and exoribonuclease region of hDIS3 are even less connected than observed in the open conformation of yeast yRrp44, with a large solvent channel between them (*Figure 5A,B*).

When inspecting the reconstructed 3D volume for un-modeled density features, we could observe a tube of density corresponding to a long single-stranded RNA bound in the channel path of hEXO-10 (*Figure 6A*). The ribonucleotide chain binds a positively-charged surface of hRRP4 and is threaded from the cap ring into the internal chamber of hEXO-9. Here, the RNA approaches the N-terminus of hMTR3 and proceeds by binding between hRRP41 and hRRP45, at the same binding path that is conserved not only in the yeast exosome (*Bonneau et al., 2009*; *Makino et al., 2013*) but even in archaeal exosome-like complexes (*Lorentzen and Conti, 2005*). After exiting from hEXO-9, the ribonucleotide chain continues directly into hDIS3. First, it binds in a surface groove formed by the juxtaposition of the PIN and CSD2 domains. The PIN domain engages a lateral surface to shape the RNA-binding cleft, while the adjacent front surface (which contains the predicted endoribonuclease site [*Lebreton et al., 2008*; *Schaeffer et al., 2009*; *Schneider et al., 2009*; *Tomecki et al., 2010*]) is exposed to solvent (*Figure 5A*). The ribonucleotide chain then flanks the CSD lobe and traverses the wide solvent-exposed channel that separates the PIN and exoribonuclease regions. The RNA continues in the funnel-like cleft between CSD1 and S1 domains and finally enters the RNB domain to reach the exoribonuclease active site. The RNA-binding residues in the RNB domain (*Figure 6D*) are conserved in all members of this protein family. The entrance into the RNB domain from the apical CSD1-S1 route is similar to bacterial RNase II and to DIS3L2 and differs from the lateral CSD1-RNB route of yeast Rrp44, consistently with the structural organization of their OB-fold domains.

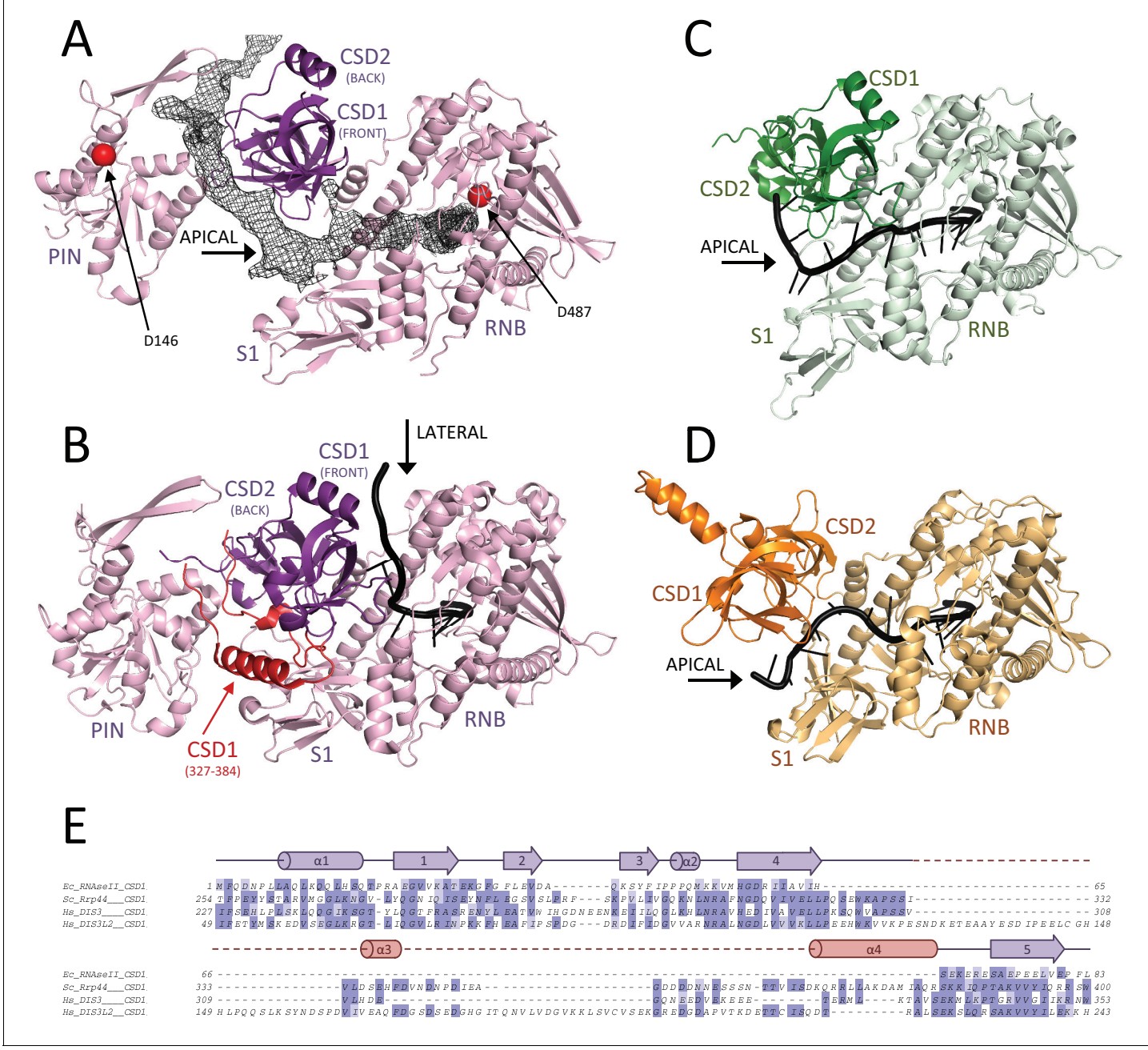

**Figure 5.** RNA binding to hDIS3 is more similar to hDIS3L2 and RNase II than to yRrp44. (**A**) The structure of hDIS3 from the cryo-EM study is shown with the density corresponding to the bound RNA, segmented from the autorefined hEXO-10 map. (**B**) The structure of the *S. cerevisiae* orthologue yRrp44 in the open conformation with an RNA molecule accessing the RNB active site through the lateral entry. CSD1 region which impairs apical entry through the CSD1-S1 route is depicted in red (*Makino et al., 2015*). (**C**) The crystal structures of the paralogue mouse DIS3L2 (*Faehnle et al., 2014*) and (**D**) of the *E. coli* RNase II (*Frazão et al., 2006*). All structures are shown in the same orientation after optimal superposition of their RNB domains. In all panels the RNB and S1 domains form a rather rigid module and are shown in lighter colors, while the CSD1 and CSD2 domains are shown in darker colors. (**E**) Sequence alignment of the CSD1 domains from *E. coli* RNase II, *S. cerevisiae* yRrp44, human DIS3, and human DIS3L2. The secondary structure of the yRrp44 CSD1 region that impairs apical entry through the CSD1-S1 route is depicted in red.

DOI: https://doi.org/10.7554/eLife.38686.012

The following figure supplement is available for figure 5:

**Figure supplement 1.** Zoomed in views of the hDIS3.

DOI: https://doi.org/10.7554/eLife.38686.013

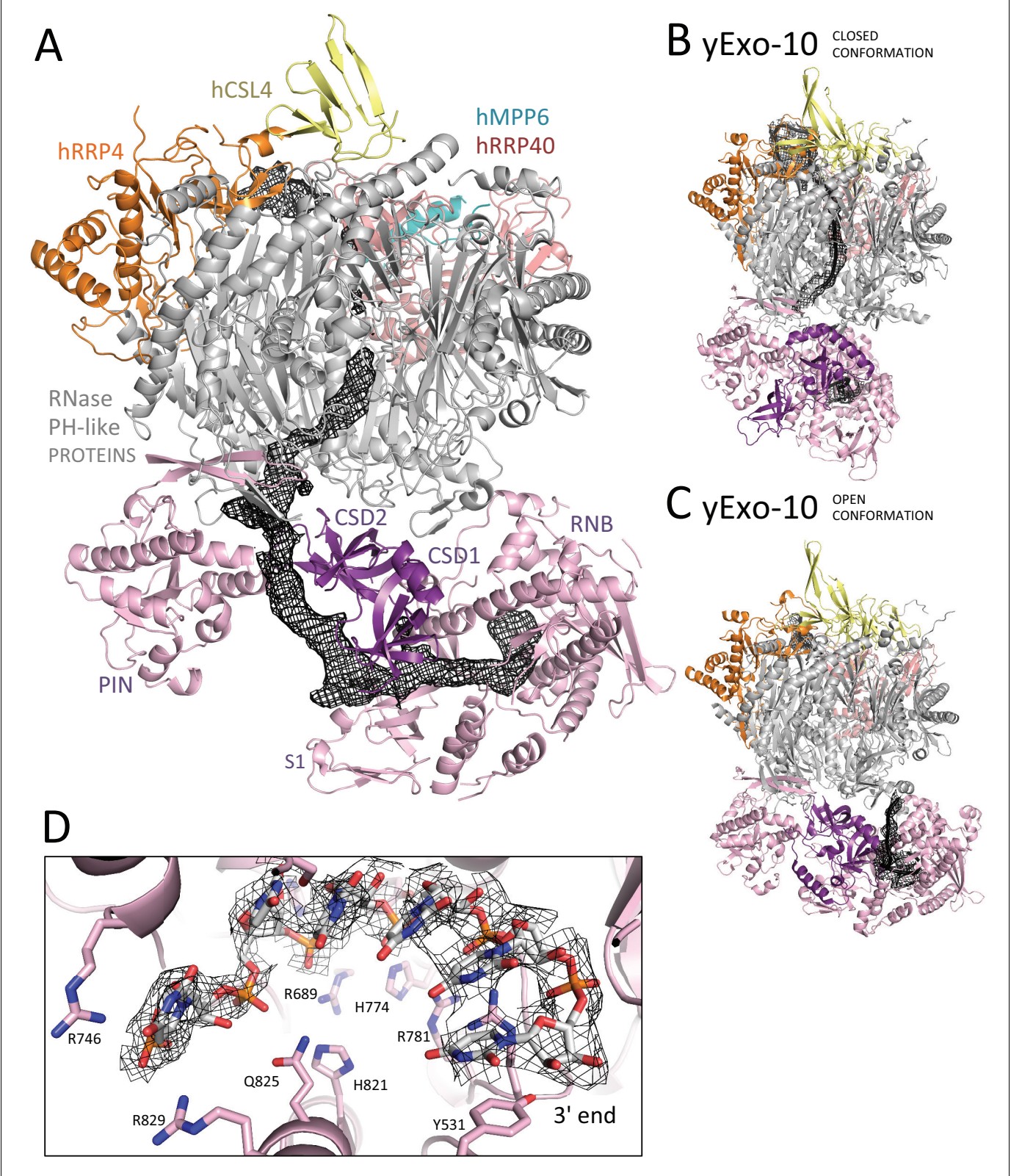

**Figure 6.** The RNA channel paths in the human and yeast exosome core complexes. (**A**) Cartoon representation of the hEXO-10$_{cat}$-hMPP6 cryo-EM structure with highlighted the density corresponding to a single-stranded RNA traversing the exosome channel, segmented from the autorefined hEXO-10 map. (**B-C**) Comparison with the two yeast exosome crystal structures with RNA bound in the channel path (**B**) (*Makino et al., 2015*), PDB: 5c0x) and in the direct path (**C**) (*Makino et al., 2015*), PDB: 5c0w). The structures are shown in the same orientation after superposition of the yExo-9

*Figure 6 continued on next page*

*Figure 6 continued*

barrels. The open and closed conformations of yRrp44 are indicated. (D) Zoom-in of at the active site of hDIS3. Meshed density from the hEXO-10 map surrounds six nucleotides built in the atomic model. Conserved residues are highlighted, and correspond to similar interactions in yRrp44 (*Lorentzen et al., 2008*; *Makino et al., 2013*).

DOI: https://doi.org/10.7554/eLife.38686.014

The long RNA channel path in the human exosome is thus formed by the combination of an open conformation of hDIS3 and an apical entry to the exoribonuclease active site. In contrast, the long RNA channel path in the yeast exosome is formed by a closed conformation of yRrp44 and a lateral entry to the exoribonuclease active site (*Bonneau et al., 2009*; *Makino et al., 2013*). In the yeast complex, the open conformation of yRrp44 is not compatible with the formation of the long RNA channel path, as a possible access to the lateral entry of Rrp44 is sterically blocked by structural features of yeast exosome subunits (yRrp43 residues 100–120, [*Zinder et al., 2016*]). The features we observed in the cryo-EM reconstruction rationalize the differences between the human and yeast core complexes that we had observed in the biochemical assays (*Figure 1*). First, the RNA channel path in the structure of the human exosome core is longer as compared to that in the *S. cerevisiae* complex (*Makino et al., 2013*), explaining the longer footprint in the RNase protection assays (*Figure 1B*). Second, the segment of RNA connecting hEXO-9 and hDIS3is is more exposed to solvent than in the closed conformation of the yeast complex, explaining the increased sensitivity to small RNases in the protection experiment (*Figure 1B*).

## The human nuclear exosome cofactors are poised at the entry of the core complex

A population of particles presented additional density corresponding to the nuclear cofactors in hEXO-14$_{cat}$. The atomic model we had built in the cryo-EM structure of hEXO-10$_{cat}$-hMPP6 could be fitted without modifications in the density of hEXO-14$_{cat}$, which featured the same open conformation of hDIS3. Although the resolution we achieved for this particle population was lower than for the exosome core, we could identify the nuclear cofactors based on their distinct structural features and knowledge from the cryo-EM reconstruction of a yeast nuclear exosome complex (*Schuller et al., 2018*). Indeed, we could fit the atomic coordinates of yMtr4-yRrp6$_N$-yRrp47 in the equivalent position on the human exosome core (*Figure 7A*). Briefly, the C-terminal helicase region of yMtr4 includes the DExH core (which carries the RNA unwinding activity) and an arch structure consisting of a stalk and a KOW domain. In the hEXO14$_{cat}$ reconstruction, density corresponding to the DExH core of hMTR4 is present on the top of hEXO-9, binding a similar surface of the cap protein hRRP4 and in a similar edge-on conformation as observed in yeast yExo-14 (*Schuller et al., 2018*). In the cryo-EM reconstruction of yeast yExo-14 bound to a pre-60S particle, the KOW domain of yMtr4 recognized a double stranded segment of the 25S rRNA substrate and the upper portion of the stalk bound yRrp6$_N$-yRrp47 (*Schuller et al., 2018*). In the reconstruction of hEXO14$_{cat}$, the KOW domain is more flexible, but ordered density is present for the stalk and the hRRP6$_N$-hRRP47 heterodimer. Careful inspection of the reconstruction corresponding to hMTR4-hRRP6$_N$-hRRP47 revealed a small density feature at the same position on the DExH core where we had previously mapped the binding of the N-terminal segment of yMpp6 (*Schuller et al., 2018*) (*Figure 7A*, left panel). Thus, the recruitment and edge-on conformation of the nuclear helicase over the entrance of the exosome core does not appear to be a specialized feature of the yeast exosome when bound to the pre-60S substrate, but rather an evolutionary conserved assembly of the nuclear exosome cofactors.

When examining the interaction between yRrp6$_N$-yRrp47 and yMtr4 in the cryo-EM structure of yeast yExo-14, we had previously noted that a conserved positively-charged surface on the yMtr4 DExH core approaches a conserved negatively-charged surface of yRrp6$_N$ (*Schuller et al., 2018*). Specifically, Arg887, Arg890 and Arg891 of yMtr4 point towards Asp86, Glu90, and Asp96 of yRrp6. The finding of similar architectural features in the positioning of hMTR4 and hRRP6$_N$-hRRP47 in the human complex (with the corresponding, conserved residues Arg856, Arg859 and Arg860 in hMTR4 and Asp118, Glu121 and Asp129 in hRRP6) prompted us to assess the importance of this conserved interaction in vivo. To this end, we went back to the yeast model system (*Figure 7B*). We had

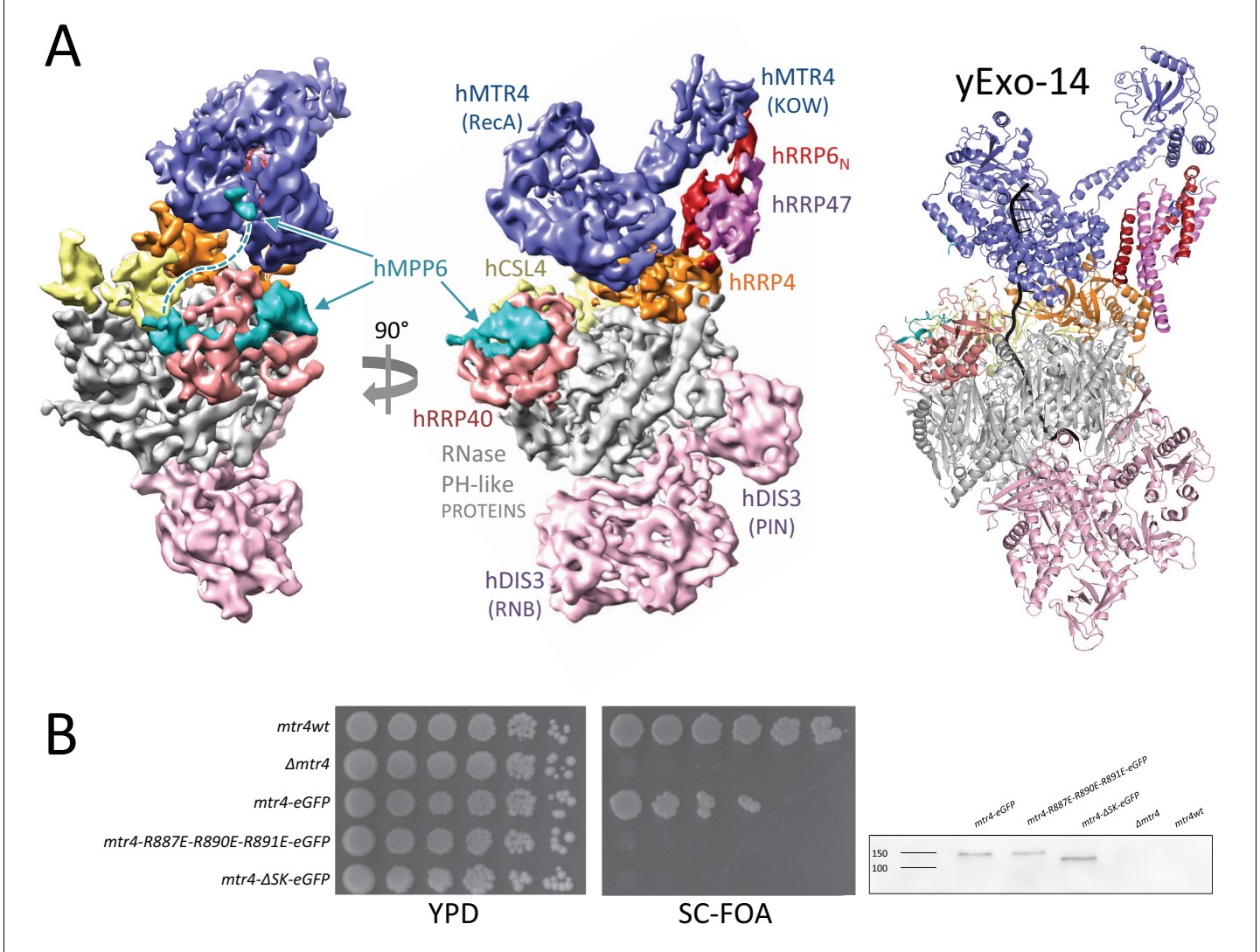

**Figure 7.** Structural conservation of human and yeast nuclear exosome cofactors. (**A**) On the left and central panels is the surface representation of the hEXO-14$_{cat}$ cryo-EM structure. On the right is a cartoon representation of the yExo-14 structure from (*Schuller et al., 2018*, PDB: 6fsz), shown in the same orientation and color coding. (**B**) On the left is a growth assay of wild-type and mutant *mtr4* strains. Endogenous *MTR4* was replaced with wild-type or mutant *mtr4-EGFP* fusions. Cells were grown to early exponential phase, and serial dilutions were spotted onto 5-fluoroorotic acid (FOA) medium or control plate. Medium containing FOA selects for the loss of the rescue vector. SC, synthetic complete medium; YPD, yeast extract peptone adenine dextrose; FOA, 5-fluoroorotic acid. ΔSK corresponds to an arch-less mutant of yMtr4 (*Falk et al., 2014*) On the right is the analysis of the expression levels of wild type yMtr4-EGFP and mutants by Western blotting using an anti-GFP antibody.

DOI: https://doi.org/10.7554/eLife.38686.015

The following source data is available for figure 7:

**Source data 1.** Detailed information of the yeast strains used in this study.

DOI: https://doi.org/10.7554/eLife.38686.016

previously integrated wild-type *MTR4* or mutant alleles as C-terminal EGFP fusions at the endogenous locus in a W303 diploid yeast strain in which one of the chromosomal copies of *MTR4* had been deleted (*Falk et al., 2014*). Using this strategy, we showed that the control *mtr4-wt-EGFP* strain was viable albeit somewhat impaired as compared to the wild-type W303 strain, while an *mtr4-ΔSK-EGFP* mutant lacking the entire arch domain (ΔSK for Δstalk-KOW) showed a severe growth defect (*Falk et al., 2014*). We used the same strategy to introduce R887E, R890E, R891E (RRR) mutations in yMtr4. Growth of the corresponding *mtr4-RRR-EGFP* strain was severely affected (*Figure 7B*), indicating that impairing the yRrp6$_N$-yRrp47-binding surface of yMtr4 has deleterious

effects in yeast. These results suggest that the conserved interaction between yMtr4 and the yRrp6$_N$-yRrp47 module is important for function in vivo.

## Conclusions

It is generally assumed that orthologous proteins and complexes sharing a high degree of sequence conservation will also share a high degree of structural and functional conservation. The yeast and human nuclear exosome complexes indeed share a similar overall structure and biochemical properties, forming macromolecular assemblies that effectively function as a cage to trap RNA substrates channeled to degradation. However, the same overall scaffold can differently modulate specific properties with subtle but impactful structural changes. One such change between the yeast and human exosome core complexes is the relative strength of the interaction between the 9-subunit scaffold and the exoribonuclease. In the case of the human complex, hDIS3 binds hEXO-9 at a similar position as the yeast complex, but with fewer interactions. The weaker binding that we detect from the structural and biochemical data rationalizes why hDIS3 was not even identified as a component in earlier proteomic studies of endogenous exosome complexes (*Chen et al., 2001*) and could only be detected with sensitive SILAC-based mass-spectrometry approaches (*Tomecki et al., 2010*). We speculate that the modulation in the strength of the interaction may have functional consequences on the regulation of hEXO-10 assembly. In yeast, there is a single yRrp44 protein that is part of both the nuclear and cytoplasmic forms of the exosome. The current view is that yeast yRrp44 assembles together with yExo-9 in the cytoplasm and is transported to the nucleus by the import capacity of yRrp6 (*Gonzales-Zubiate et al., 2017*). Instead, the situation is more complex in human cells, as there are instead two different paralogues (hDIS3 and hDIS3L) that have to be selectively incorporated in the nuclear and cytoplasmic forms of the complex (*Tomecki et al., 2010*). It is unclear at the moment where the human nuclear exosome core is assembled, but we note that bioinformatics analyses predict the presence of a possible NLS in the sequence of hDIS3, within a C-terminal part of the molecule (residues 949–958) that in the cryo-EM structure is accessibly exposed to the solvent. We speculate that a separate nuclear import of hDIS3, independent from that of hEXO-9-hRRP6-hRRP47, would allow the formation of the correct nuclear assembly, and a lower affinity of hDIS3 for hEXO-9 may be important in this context. As a note, another isoform of hDIS3 is translated from an mRNA with an alternative exon in the 5' coding region, resulting in a smaller PIN domain that might therefore further impact on the binding affinity for hEXO-9 and/or the RNA binding properties.

Unexpectedly when considering the levels of sequence conservation, the exoribonuclease region of hDIS3 is more similar in terms of RNA path to bacterial RNase II and to human DIS3L2 (a paralogue that functions independently of the exosome complex) than it is to its yeast orthologue yRrp44. The difference in turn impacts on the RNA-channeling path in the exosome core. In the yeast complex, Rrp44 can adopt either a closed conformation to support the RNA channel path or an open conformation to support a short direct path to the exoribonuclease site (*Makino et al., 2013*, *2015*). In the human complex, the RNA channel path is achieved by an open conformation of hDIS3. None of the 2D or 3D particle classes obtained during processing of the cryo-EM dataset showed a density of hDIS3 corresponding to the closed yRrp44 conformation. In principle, the open conformation of hDIS3 could also support the direct access route, but whether this may be favored for at least a subset of nuclear transcripts as in the case of yeast (*Han and van Hoof, 2016*) is currently unclear. The channel path of hEXO-10 is used also in the context of hEXO-14 by RNA substrates that are unwound by hMTR4. This helicase is positioned on top of hEXO-10 with a similar edge-on conformation as previously observed in yeast (*Schuller et al., 2018*). We note that while this work was in revision, similar structural findings were reported (*Weick et al., 2018*). Understanding the basis for the versatility of human hMTR4 in interacting with different cofactors and RNA exosome substrates is a quest for future studies.

## Materials and methods

### Mammalian cell protein expression and purification

For mammalian cell expression, stable pool generation was based on the piggyBac system (*Li et al., 2013*). Briefly, hRRP6 was cloned into a PB-T vector with an engineered Twin-strep tag (IBA)

followed by a 3C protease cleavage site and hRRP47 was cloned into a PB-T without tags. Suspension-adapted HEK293T cells were co-transfected with the PB-T plasmids, a PB-RN (reverse tetracycline trans-activator) plasmid and a pCMV-hyPBase plasmid carrying the hyperactive piggyBac transposase (Yusa et al., 2011). Cells were grown in FreeStyle293 medium (Thermo Fisher Scientific) and passaged every 2–3 days. Stable integrants were selected for 17–20 days with 500 µg/ml G418 and 10 µg/ml puromycin. Expression in stable pools was induced for 2–3 days with 10 µg/ml doxycycline at a cell density of $10^6$/ml.

Pellets of ~$5 \times 10^8$ cells were resuspended in 25 mL hypotonic buffer composed of 100 mM Tris-HCl pH 8.0, 10 mM NaCl, 1 mM EDTA, 2 mM DTT, Complete protease inhibitor (Roche), and 15 U/ml benzonase. Following 10 min incubation on ice cells were open with 10 strokes of a Dounce glass homogenizer and the nuclei were pelleted by centrifugation at 3000 rpm for 10 min 4°C. The supernatant was then supplemented with 150 mM NaCl and loaded on a 1 mL StrepTactin XT HighCapacity column (IBA). Following 50 column volumes wash proteins were eluted with 100 mM Tris pH 8.0, 150 mM NaCl, 1 mM EDTA, 2 mM DTT, and 50 mM biotin. The hRRP6(D371N)-hRRP47 dimer obtained from a single strep-affinity step was directly used to reconstitute the exosome complex.

## Bacterial protein expression and purification

All other proteins were recombinantly expressed in BL21 Star (DE3) pRARE *E. coli* cells grown at 37°C in TB media up to $OD_{600}$1.0–1.5, and induced with 0.5 mM IPTG for overnight expression at 18°C. The full-length hMTR4 wild type was expressed with an N-terminal 6xHis-GST-3C tag. The full-length hDIS3 double inactive mutant (D146N, D487N) was expressed with an N-terminal 10xHis-3C tag. The full-length hEXO-9 subunits were expressed with an N-terminal 10xHis-3C tag, except for hRRP43 and hRRP46 which were expressed with an N-terminal 6xHis-SUMO tag. Both fusion proteins used were designed with glycine-serine linkers: 10xHis-3C-hRRP4-[AS-5xGS]-hRRP6$_N$(1-160), 10xHis-3C-hMPP6-[9xGS]-hRRP40. The full-length hRRP47 was expressed with an N-terminal 6xHis-GST-3C tag. All three types of the hEXO-9 complexes: hEXO-9-WT, hEXO-9-hMPP6, and hEXO-9-hMPP6-hRRP6$_N$-hRRP47 were reconstituted with individual proteins and subcomplexes, adapting the strategy reported by Liu et al. (2006). Namely hRRP41 and hRRP45 were co-expressed and purified as a dimer, while hMTR3 and hRRP42 were co-expressed and later co-lysed with hRRP43 and purified as a trimer. The hRRP4-hRRP6$_N$(1-160) fusion was co-expressed and purified with hRRP47. Bacteria were lysed by sonication in 50 mM Tris-HCl pH 7.5, 150 mM NaCl, 10% (v/v) glycerol, 5 mM β-mercaptoethanol, 0.5 mM AEBSF, and 15 U/ml benzonase. All proteins were purified by nickel-affinity chromatography using either HisTrap HP column (GE Healthcare) or HIS-Select resin (Sigma-Aldrich). Affinity tags were cleaved with His-3C protease or His-Senp2 SUMO protease, and later removed in a second nickel-affinity step. Proteins were then subjected to ion exchange chromatography on a HiTrap Heparin HP column (GE Healthcare), except for hRRP41-hRRP45 dimer and hMTR3-hRRP42-hRRP43 trimer, which were purified over a HiTrapQ HP column (GE Healthcare). Degradation products of the hDIS3$_{cat}$ were bound to the HiTrapQ HP column while the full-length protein was recovered from the flow through. In the final step all single proteins and subcomplexes were subjected to size exclusion chromatography on Superdex 200 or 75 Increase columns (GE Healthcare) in 20 mM Hepes-NaOH pH 7.5, 150 mM NaCl, 2 mM DTT.

## Complex reconstitution and cryo-EM grid preparation

The hEXO-9-hMPP6-hRRP6N-hRRP47 complex was mixed in equimolar amount with hMTR4 and hDIS3 inactive mutant in a buffer containing 20 mM Hepes-NaOH pH 7.5, 150 mM NaCl, 2 mM MgCl2, 1 mM ADP, and 2 mM DTT to form the hEXO-14 complex. Following 30 min incubation on ice the single stranded 44-uracil RNA ($U_{44}$) was added in 1.2 molar excess. The complex was cross-linked in-batch for 30 min at room temperature with 1 mM BS3, a lysine-specific crosslinking agent (Thermo Scientific). Following quenching with 20 mM $(NH_4)_2CO_3$ the sample was applied on a Superose 6 Increase analytical column (GE Healthcare). Four microliters of the hEXO-14-RNA sample at 0.28 mg/mL were applied to glow-discharged R2/1 200 mesh holey carbon grids (Quantifoil) and immediately blotted for 3.5 s at ~95% humidity and 4°C, then plunge-frozen into liquid ethane cooled by liquid nitrogen using a Vitrobot Mark IV (FEI).

## Cryo-EM data collection and processing

We collected 8047 micrographs on a Titan Krios electron microscope (FEI) operated at 300 kV, equipped with a K2 Summit direct electron detector (Gatan) and a GIF quantum energy filter (20 e⁻V) (Gatan), and operated in electron counting mode (pixel size: 1.35 Å per pixel). Each micrograph was exposed for 10 s with a dose rate of 4.69 $e^-/Å^2/s$ (total specimen dose, 46.9 $e^-/Å^2$), and 40 frames were captured per micrograph. The SerialEM software package was used for automated-acquisition with defocus values varying from 0.5 μm to 3.5 μm. The dose-fractionated movies were gain normalized, aligned and dose-weighted using MotionCorr2 (*Zheng et al., 2017*). Defocus values were estimated using GCTF (*Zhang, 2016*) and particles were automatically picked using Gautomatch. More than 690,000 particles were selected following 2D classification in RELION 2.1 (*Scheres, 2016*) to remove clear non-particle candidates (ice-contaminations, carbon-edges). Two hundred particles from several different 2D classes in multiple orientations were used to generate an ab initio model. 3D Classification using six classes yielded two distinct molecular assemblies, one showing the hEXO-10 the other the hEXO-14 complex that were subsequently aligned and classified separately. 3D Refinement of the hEXO-10 particle population yielded an overall resolution of ~3.80 Å, while the hEXO-14 complex was refined to a global resolution of ~6.25 Å. All global resolutions were estimated by applying a soft mask around the protein density and using the gold standard Fourier shell correction (FSC) = 0.143 criterion, as implemented in the RELION post-processing routine. Both maps were carefully interpreted in their respective resolution scheme. The higher resolved hEXO-10 complex displayed near-atomic resolution information and thus it was possible to refine and analyze side-chain positions. The atomic model of the hEXO-10 structure was built in the cryoEM density map using COOT (*Emsley et al., 2010*) and refined with Phenix (*Afonine et al., 2018*). For the lower resolved hEXO-14 complex the individual protein domains were fitted independently as rigid bodies.

## In vitro assays

Body-labeled RNAs were generated by in vitro transcription with the MEGAshortscript transcription kit (Ambion) in presence of [α-$^{32}$P] UTP (Perkin-Elmer) and RNase T1 (ThermoFisher), to remove leading guanosines, followed by denaturing gel purification. Templates were obtained by annealing of two DNA oligonucleotides containing the T7 promoter sequence. The final sequence for the 97-mer was (CU)$_{48}$C. Proteins (5 pmol each) were mixed with 2.5 pmol $^{32}$P body-labeled RNA to a final 10 μl reaction volume in 50 mM HEPES-NaOH (pH 7.5), 50 mM NaCl, 5 mM magnesium diacetate, 10% (w/v) glycerol, 0.1% (w/v) NP40, and 1 mM DTT. After incubation for 45 min at 4°C, reactions mixtures were treated with 0.5 μg RNase A and 1.25 U RNase T1 (Fermentas) or with Benzonase (endonuclease from *Serratia marcescens), 375 U* for 20 min at 25°C. Protected RNA fragments were then extracted twice with phenol:chloroform:isoamyl alcohol (25:24:1, v/v, Invitrogen), precipitated with ethanol, separated on 12% (w/v) denaturing PAGE, and visualized by phosphorimaging (Fuji).

## Yeast strains

All yeast strains were based on W303 MATa/MATα {leu2-3,112 trp1-1 can1-100 ura3-1 ade2-1 his3-11,15, RAD5}, as previously described in (*Falk et al., 2014*). For detailed list see *Figure 7—source data 1*. Yeast cells were grown to OD$_{600}$ ~1 AU. 1 mL was then harvested, washed once in ddH2O, serially diluted 1:5 and spotted on non-selective (YPD) and selective plates (SC-FOA, SCØURA, YPD/G418). Cells were incubated for 3 days at 30°C. Western blot analyses were made with α-GFP-mouse-mAb and goat-α-mouse HRP mAb.

## Acknowledgements

We would like to thank Elisabeth Stegmann and Judith Ebert for support in protein purification, Daniela Wartini for mammalian cells expression, Alex Kögel for the initial reconstitution of the hEXO-10, Mike Strauss for giving superb assistance and training to access the microscopes of the MPIB cryo-EM Facility, members of our group for comments and discussions. This study was supported by an EMBO Long-Term Fellowship (ALTF 1008–2015) to PG. and the Max Planck Gesellschaft, the European Commission (ERC Advanced Investigator Grant EXORICO), the Deutsche Forschungsgemeinschaft (DFG SFB646, SFB1035, GRK1721), and the Louis Jeantet Foundation to EC. The cryo-EM

density maps are deposited in the Electron Microscopy Data Bank under accession numbers EMD-0127 and EMD-0128. The atomic model is deposited in the Protein Data Bank (PDB) under accession number 6H25. The authors declare no competing financial interests.

## Additional information

### Funding

| Funder | Grant reference number | Author |
|---|---|---|
| European Molecular Biology Organization | ALTF 1008-2015 | Piotr Gerlach |
| European Commission | ERC-2016-ADG 740329 EXORICO | Elena Conti |
| Deutsche Forschungsge-meinschaft | SFB646 | Elena Conti |
| Louis-Jeantet Foundation | | Elena Conti |
| Max-Planck-Gesellschaft | | Elena Conti |
| Deutsche Forschungsge-meinschaft | SFB1035 | Elena Conti |
| Deutsche Forschungsge-meinschaft | GRK1721 | Elena Conti |

The funders had no role in study design, data collection and interpretation, or the decision to submit the work for publication.

### Author contributions
Piotr Gerlach, Reconstituted the recombinant human complexes, Collected and processed cryo-EM data, Wrote the manuscript; Jan M Schuller, Collected and processed cryo-EM data, Wrote the manuscript; Fabien Bonneau, Carried out the RNase protection assays and established the stable cell lines; Jérôme Basquin, Built and refined the atomic model; Peter Reichelt, Carried out the yeast experiments; Sebastian Falk, Purified yeast complexes and revised the manuscript; Elena Conti, Designed and managed the project, Wrote the manuscript

### Author ORCIDs
Piotr Gerlach (ID) http://orcid.org/0000-0001-9599-7322
Jan M Schuller (ID) https://orcid.org/0000-0002-9121-1764
Fabien Bonneau (ID) https://orcid.org/0000-0001-8787-7662
Sebastian Falk (ID) http://orcid.org/0000-0001-7848-4621
Elena Conti (ID) http://orcid.org/0000-0003-1254-5588

### Decision letter and Author response
Decision letter https://doi.org/10.7554/eLife.38686.025
Author response https://doi.org/10.7554/eLife.38686.026

## Additional files

### Supplementary files
• Transparent reporting form
DOI: https://doi.org/10.7554/eLife.38686.017

### Data availability
The cryo-EM density maps are deposited in the Electron Microscopy Data Bank under accession numbers EMD-0127 and EMD-0128. The atomic model is deposited in the Protein Data Bank (PDB) under accession number 6H25.

The following datasets were generated:

| Author(s) | Year | Dataset title | Dataset URL | Database, license, and accessibility information |
|---|---|---|---|---|
| Schuller JM, Falk S, Basquin J, Conti E | 2018 | Human nuclear RNA exosome EXO-14 complex (cryo-EM density map) | www.ebi.ac.uk/pdbe/en-try/emdb/EMD-0127 | Publicly available at the Electron Microscopy Data Bank (accession no: EMD-0127) |
| Gerlach P, Schuller JM, Falk S, Basquin J, Conti E | 2018 | Human nuclear RNA exosome EXO-10-MPP6 complex (cryo-EM density map) | www.ebi.ac.uk/pdbe/en-try/emdb/EMD-0128 | Publicly available at the Electron Microscopy Data Bank (accession no: EMD-0128) |
| Gerlach P, Schuller JM, Falk S, Basquin J, Conti E | 2018 | Human nuclear RNA exosome EXO-10-MPP6 complex (atomic model) | www.rcsb.org/structure/6H25 | Publicly available at the RCSB Protein Data Bank (accession no: 6H25) |

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
