## [Decision Letter]

Thank you for submitting your article "Distinct and evolutionary conserved structural features of the human nuclear exosome complex" for consideration by *eLife*. Your article has been favorably reviewed by two peer reviewers, including Sjors HW Scheres as the Reviewing Editor and Reviewer #1, and the evaluation has been overseen by John Kuriyan as the Senior Editor. The following individual involved in review of your submission has agreed to reveal their identity: David Tollervey (Reviewer #2).

The reviewers have discussed the reviews with one another and the Reviewing Editor has drafted this decision to help you prepare a revised submission. Note that the revisions do not require any new experimental work.

Summary:

The authors report the structural analysis of the human nuclear exosome core alone and in complex with key cofactors. The structure shows strong similarities to previous yeast work as might be expected, but also reveals significant differences that underpin functional features, e.g. the reduced affinity of Dis3 for the human core complex. Overall, this is a good piece of structural biology that gives insights into the structure and functional interactions of an important complex. A few points might have been more extensively discussed, these are listed below.

Essential revisions:

1) Subsection “The human nuclear exosome cofactors extend the RNA channel of the core Complex”: (Figure 1C)

Mtr4 without other exosome cofactors does not confer clear protection (lane 3). Lane 4 (Exo10-Rrp6-Rrp47 – Mtr4) and 7 (Exo10-Mpp6 – Mtr4) show the same profile with a modest shift towards longer species (around 50nt) and a stronger shift when all the cofactors are present (Lane 8). It would be useful to discuss the apparent finding that the Mtr4-exosome interaction needs at least one cofactor to be stabilized and all of them to show strong association.

Longer fragments (around 50nt) are stabilized when there is no ribonuclease domain of Rrp6 (lane 10 compared to lane 9). Is this because of the competition between Mtr4 and Rrp6 bound on same site on top of the exosome? It would be useful to discuss why such "competition" would prevent protection of longer fragments in human but not in yeast (comparing Figure 1C, lane 8 with Figure 1D, lane 4)?

2) Subsection “The human nuclear exosome cofactors are poised at the entry of the core Complex”:

The growth test is consistent with previous studies showing that Mtr4 is required for Rrp6 activity, indicating that the interaction of N-terminal domain of Rrp6 with Mtr4 is functionally important. The structure presented in (Shuller et al., 2018) shows that binding by Rrp6 and Mtr4 to the top of the exosome are mutually exclusive. What would be the potential structure with both full length Rrp6 and Mtr4?

3) It is hard to assess how good the cryo-EM density is from Figure 2 and Figure 7. It would be useful if the authors could provide figure supplements to Figure 3, (perhaps also Figure 4), Figure 5 and Figure 7 with zoomed in views of some α-helices, β-strands and/or stretches of main chain with resolved side-chain density as an illustration of how good the experimental density is the relevant parts of the map.

4) In Figure 2—Figure supplement 3D, the authors show a map-vs-model FSC curve in green. This curve shows an excellent correlation between the model and the map, all the way up to the resolution of the map. This is a good thing. However, the authors should describe how they prevented overfitting of the model inside the cryo-EM density, as at these resolutions, by using a stronger weight on the map, the FSC curve could have been artificially made higher. We suggest that the authors test different weights in the model refinement program by refining the model in one of the half-set reconstructions, and then calculate 2 FSC curves for each weight: FSC_work between the refined model and the half-set used for that refinement, and FSC_test between the same model and the other half-set. For too high weights, the FSC_work and FSC_test curves will start to become very different as a result of overfitting. By keeping the weight on the density lower, this overfitting can be limited. The final model refinement should then be performed using the thus determined weight, and the corresponding FSC model-vs-map curve should be shown.

5) Were COOT and Phenix, which were used for the crystal structure determination, also used for the cryo-EM modelling? If so, this should be mentioned in the Materials and methods section.

---

## [Author Response]

Essential revisions:1) Subsection “The human nuclear exosome cofactors extend the RNA channel of the core Complex”: (Figure 1C)Mtr4 without other exosome cofactors does not confer clear protection (lane 3). Lane 4 (Exo10-Rrp6-Rrp47 – Mtr4) and 7 (Exo10-Mpp6 – Mtr4) show the same profile with a modest shift towards longer species (around 50nt) and a stronger shift when all the cofactors are present (lane 8). It would be useful to discuss the apparent finding that the Mtr4-exosome interaction needs at least one cofactor to be stabilized and all of them to show strong association.

We thank the reviewer. We have added this discussion in the text:

“Incubating hEXO-10_cat_ and hMTR4 in the presence of either hRRP6_cat_-hRRP47 (Figure 1C in lane 4) or hMPP6 (lane 7) showed only a modest shift to longer species (~50 nucleotides). The presence of all four cofactors and ADP was required to have a stronger shift to ~50 nucleotides (lane 8). Thus, it appears that the Mtr4-exosome needs at least one cofactor to be stabilized and all of them to show strong association, both in the yeast and human system.”

Longer fragments (around 50nt) are stabilized when there is no ribonuclease domain of Rrp6 (lane 10 compared to lane 9). Is this because of the competition between Mtr4 and Rrp6 bound on same site on top of the exosome? It would be useful to discuss why such "competition" would prevent protection of longer fragments in human but not in yeast (comparing Figure 1C, lane 8 with Figure 1D, lane 4)?

We believe this is probably due to the poor behavior of human full-length RRP6 as compared to yeast full-length Rrp6.

We have added the sentence: “Using covalently linked hRRP6_N_ instead of f.l. hRRP6 appeared to even stabilize the 50 nucleotide fragments (compare lanes 8 and 10). The most likely reason for this stabilization is the improved biochemical properties and stability of hRRP6_N_ as compared to the full-length protein, allowing us to overcome the hRRP6 stoichiometry issues in the reconstituted complexes (Figure 1A).”

2) Subsection “The human nuclear exosome cofactors are poised at the entry of the core Complex”:The growth test is consistent with previous studies showing that Mtr4 is required for Rrp6 activity, indicating that the interaction of N-terminal domain of Rrp6 with Mtr4 is functionally important. The structure presented in (Shuller et al., 2018) shows that binding by Rrp6 and Mtr4 to the top of the exosome are mutually exclusive. What would be the potential structure with both full length Rrp6 and Mtr4?

We would expect the ribonuclease domain of hRRP6 would relocate to a different position on the exosome. In yeast, it binds to a position located on the side on the side. However, in the cryo-EM structure of the yeast complex, the density is not well defined, suggesting flexibility.

3) It is hard to assess how good the cryo-EM density is from Figure 2 and Figure 7. It would be useful if the authors could provide figure supplements to Figure 3, (perhaps also Figure 4), Figure 5 and Figure 7 with zoomed in views of some α-helices, β-strands and/or stretches of main chain with resolved side-chain density as an illustration of how good the experimental density is the relevant parts of the map.

We now included Figure 3—figure supplement 2 and Figure 5—figure supplement 1 with views of secondary structure elements in representative parts of the hEXO-10 structure, showing resolved side chain density. As discussed in the text, the density of Exo-14 is at 6.25 Å resolution, and therefore the side chains are not resolved.

4) In Figure 2—figure supplement 3D, the authors show a map-vs-model FSC curve in green. This curve shows an excellent correlation between the model and the map, all the way up to the resolution of the map. This is a good thing. However, the authors should describe how they prevented overfitting of the model inside the cryo-EM density, as at these resolutions, by using a stronger weight on the map, the FSC curve could have been artificially made higher. We suggest that the authors test different weights in the model refinement program by refining the model in one of the half-set reconstructions, and then calculate 2 FSC curves for each weight: FSC_work between the refined model and the half-set used for that refinement, and FSC_test between the same model and the other half-set. For too high weights, the FSC_work and FSC_test curves will start to become very different as a result of overfitting. By keeping the weight on the density lower, this overfitting can be limited. The final model refinement should then be performed using the thus determined weight, and the corresponding FSC model-vs-map curve should be shown.

We performed the test suggested by the reviewer and include the results Figure 2—figure supplement 3E, arguing that our model is not over-fitted. We validated the model by displacing the atom positions in the final model on average by 0.5 Å, followed by real space refinement using phenix.real_space_refine against the first of the two independent half maps (FSC_work). The resulting refined model was then used to calculate Model-Map FSCs against the second half-map (FSC_test) that was not used for refinement. The FSC_work and FSC_test curve show a very good agreement in the entire resolution range, validating the entire structure for over fitting.

5) Were COOT and Phenix, which were used for the crystal structure determination, also used for the cryo-EM modelling? If so, this should be mentioned in the Materials and methods section.

Included.